# Influencing Factors of Traditional Village Protection and Development from the Perspective of Resilience Theory

Xiangbin Wang [†] and Quan Zhu *,[†]

School of Landscape and Architecture, Zhejiang Agriculture and Forestry University, Hangzhou 311300, China
* Correspondence: zhuquan@zafu.edu.cn
† These authors contributed equally to this work.

**Abstract:** In the process of urbanization in China, traditional villages are facing many challenges and impacts which impose certain constraints on their sustainable development. Based on the perspective of resilience theory, this paper uses fuzzy-set qualitative comparative analysis to construct a complex causal model of traditional village conservation and development, conducts an empirical study on 28 national traditional villages in Liandu District and Qingyuan County, Lishui City, Zhejiang Province, and analyzes four high-level history paths by using fsQCA3.0 software (Developed by the American scholar Charles Ragin et al.), based on which four traditional conservation and development models are proposed and concretized into different conservation and development strategies for different village types. The research analysis shows that (1) resilience theory is applicable to the study of traditional village conservation and development, and has certain relevance and can effectively improve the response ability of traditional villages to uncertainty risks; (2) The framework of traditional village protection and development is based on landscape environmental protection, guaranteed by village industry development, and supported by other influencing factors (natural disaster response, social organization, and cultural heritage); (3) under the perspective of resilience theory, there are four models of traditional village conservation and development (JSC model, Z&C model, J&W model, and S&C model), which have good explanatory power and coverage and can address the real problems of different village types in a targeted manner.

**Keywords:** resilience theory; traditional villages; fuzzy-set QCA; configuration path; influence factor

## 1. Introduction

Traditional villages are a reflection of the wisdom of the Chinese nation and its long history of agricultural civilizations. They are the concentrated expression of local traditional cultures, architectures, aesthetics, and spatial patterns; they are tangible representations of the harmonious relationship between the village and the surrounding natural environment. However, the past 40 years of urbanization have created conflict between traditional villages and modernity. The production, lifestyle, and cultural values of village societies have changed even to the point of deconstruction. A governmental system of the past, loosely translated as "focusing on the city and not the countryside", diverted more development resources to cities, causing traditional villages to lag behind in terms of social economies, public services, infrastructures, and other aspects. Today, there is a significant gap between urban and rural areas.

In 2012, the Ministry of Housing and Urban–Rural Development announced its fifth set of Chinese traditional village lists. A total of 6819 villages were included in the list, covering all provinces in the country except Hong Kong, Macao, and Taiwan. Protected Chinese traditional villages only account for a portion of the large number of villages in China, most of which are in danger of disappearing [1]. The risks and disturbances faced by traditional villages include threats to their aesthetics and spatial patterns, social networks, local customs, structures, and landscapes; degradation of the physical environment of a

traditional village is also an existential threat to its culture, history, and memory. In the context of modern urbanization, it is crucial to protect traditional villages in the process of their continued development.

The present study was conducted to explore the protection and development of traditional villages from the perspective of resilience theory, to investigate the relationship between resilience theory and the protection and development of traditional villages, and to construct a relational framework that reflects its internal mechanism. QCA (Qualitative Comparative Analysis) was used to analyze the influencing factors of the protection and development of traditional villages from the perspective of resilience. The core configuration paths that affect the protection and development of traditional villages were determined accordingly. The results of this work may improve the ability of traditional villages to resist risks and shocks; they may also provide a workable reference for the protection and development of traditional villages in other regions of our country.

## 2. Theoretical Background and Relevant Research

Resilience is a necessary quality for cities and villages around the world to maintain sustainable social, economic, cultural, ecological, and other forms of development [2]. Resilience reflects the capability of a system to maintain its basic structural characteristics after it is disturbed, including maintaining a dynamic balance through self-organization, learning, and updating its operations. Over the years, the concept of urban resilience has undergone multiple transformations and applications from engineering resilience to evolutionary resilience, forming a relatively complete theoretical system. Compared with cities, rural resilience has received less research attention.

As an important part of the rural regional system, challenges faced by traditional villages currently include lack of infrastructure, lack of endogenous development momentum, weak grass-roots organization construction capabilities, and lack of capital, talent, and/or technology. Villages are more vulnerable to uncertainties than cities. Practices grounded in resilience theory may help to soundly guide the protection and development of traditional villages [3] (Figure 1).

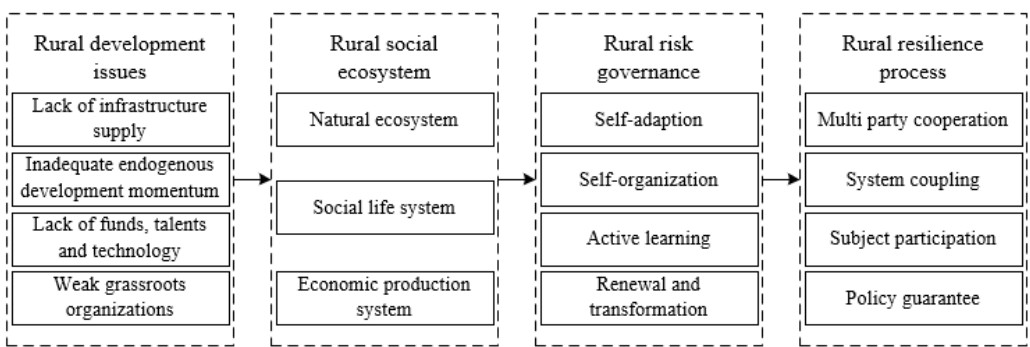

**Figure 1.** Logical framework of rural resilience (data source: reference [3]).

### 2.1. Resilience Theory and Its Application

Since the beginning of the 21st century, scholars at home and abroad have begun to apply resilience theory in rural areas. Contextually, rural resilience theory is characterized by dynamic balance, emphasizing that the countryside is a self-healing system [2,4] with the ability to resist changes, absorb disturbances, and adapt under uncertain circumstances. Positive responses maintain effective, i.e., resilient, development [3,5–7]. Wilson further enriched the definition of rural resilience on this basis, arguing that rural resilience is a process of continuously maintaining the balance between rural social, economic, and environmental needs while successfully withstanding endogenous and exogenous changes [8]. Theoretically, research on rural resilience mostly exists in the fields of geography and sociology; scholars of different disciplines may differ significantly in their definition of this concept. For example, geographers [9] believe that rural resilience includes the ability of

rural systems to resist and adapt to external disturbances and shocks, as well as the ability to transform and achieve new development [10]. Sociologists regard rural resilience as a governance method adopted by multiple subjects to maintain organizational stability and promote organizational change in response to risks [11].

According to Wang Xiangbin's review of international research, rural resilience is the ability of a rural system to maintain sustainable development after it is negatively impacted by adverse factors. This resilience in turn increases the stability and sustainability of the development process. Different scholars have different focuses in terms of their application, mainly focusing on rural ecological resilience (natural disaster response, landscape protection), social resilience (social network structure, social governance model, inheritance of cultural elements), economic resilience (traditional agriculture) transformation, new economic integration), and other areas [3] (Table 1).

**Table 1.** Domestic and foreign rural resilience application fields (data source: reference [3]).

| Research Contents | Research Priorities |
|---|---|
| **Ecological resilience** (natural disaster response, landscape protection) | Evaluate the resilience of villages in disaster response, climate change, environmental landscape, and other aspects; use engineering and non-engineering measures to improve the ability of villages to cope with natural ecological changes |
| **Social resilience** (social network structure, social governance model, cultural element inheritance) | Based on the evolution mechanism of rural resilience, analyze the impact of rural governance, rural policies, and management measures on rural resilience and the relationship with sustainable development in rural areas |
| **Economic resilience** (transformation of traditional agriculture and integration of new economy) | Based on the structural dynamics equation, livelihood trajectory, and other methods, determine the roles and impacts of economic factors such as agricultural basic resources, rural livelihoods, and industrial development on rural resilience |
| **Economic resilience** (transformation of traditional agriculture, integration of new economy) | Exploring the role and impact of economic factors such as basic agricultural resources, rural livelihoods, and industrial development on rural resilience through the use of structural dynamics equations, livelihood trajectories, and other methods. |

(1) **Natural ecology field**. This field of rural resilience research mainly includes two aspects of resilience: response to natural disasters and repair and restoration of the landscape environment, and there exist two more common approaches: engineering and non-engineering measures. ① Engineering measures show that physical construction or engineering techniques can effectively reduce potential impacts and improve hazard resistance. Ling, Zijian, and other scholars have also proposed an enhanced focus regarding rural disaster prevention and control in their natural disaster response studies, encouraging rural villages to integrate institutional, social, and engineering dimensions for disaster prevention and mitigation construction [12]. In the study of a rural watershed environment in the southern Jiangsu water network, Ding Jinhua constructed a multidimensional watershed environment resilience system from three perspectives: structural resilience, technical resilience, and process resilience [13]. ② Non-engineering measures have a complementary role in rural resilience capacity building by actively promoting disaster prevention and mitigation, and natural ecological knowledge, to enhance local residents' awareness of environmental protection as well as disaster resilience and relief capabilities. Chen et al. used coastal rural areas

in Taiwan, China, as an example, and used in-depth interviews to help local residents learn disaster prevention skills, and the results showed that they could help local people effectively mitigate the effects of disasters [14].

(2) **Social life field**. Rural governance and rural policy are often the focus of research on rural resilience, and both can be seen as a way for different rural subjects to cope with disturbances. (1) Rural governance. Scholars such as Holling have conducted continuous research on resilience assessment, mechanism evolution, resource allocation, and resilience enhancement, emphasizing that rural systems are constantly renewed in four stages of development, protection, release, and renewal in the face of external shocks [6], and focusing on adaptive management approaches to enable good rural governance [15]. Tang Renwu et al. analyzed the internal logic and mechanism of rural governance from the perspective of evolutionary resilience, clarified the main body of rural governance, and proposed suggestions to enhance resilient governance capacity and thus promote rural revitalization [16]. (2) Rural policy is an important guarantee for improving rural resilience, and scholars such as Folke encourage the incorporation of resilience criteria into policy formulation, emphasize the interrelationship between social–ecological systems, and increase policy diversity to cope with uncertainty [17] and changes in resilience from policy adoption [18]. Therefore, rural governance and policy formulation should pay attention to the adequacy and foresight in dealing with rural issues and form a bottom-up resilience management system so that it can be more effective in the rural economy and society.

(3) **Economic production field**. Resilience research in the field of rural economic production mainly includes two aspects, such as industrial development and rural livelihood. Among them: ① Agriculture is the most basic form of industry and economy in the countryside, which is related to the sustainable development of the countryside to a certain extent. However, if villages rely too much on a single form of agricultural production, their overall level of resilience will continue to decrease. Resilience theory emphasizes the diversification of the composition of the rural economic production system, promotes the transformation of traditional agriculture to modern agriculture, and introduces cultural industries, tourism and other special industries. Therefore, strengthening the development of rural industries is of great significance for the construction of rural resilience, and scholars such as Yu Wei attempted to conduct special studies on the evaluation of agricultural development resilience, the evolution of spatial and temporal patterns and influencing factors, in order to promote the transformation and development of agricultural industries and improve the overall development pattern of the region [19]. (2) Directly related to rural industries are rural livelihoods; Sallu et al. adopted a livelihood trajectory approach to explore shocks and stresses affecting rural livelihoods and to elucidate livelihood strategies that help enhance resilience [20]. Zou Yu et al. used the Probit–ISM model to analyze the influencing factors of nonfarm employment, based on the perspective of livelihood resilience, to provide a reference for decision making to consolidate the effectiveness of poverty eradication [21]. Therefore, enhancing the resilience and vitality of rural systems should pay special attention to the transformation and development of traditional agriculture, as well as the economic integration of new industries, to drive rural economic production efficiency and villagers' income, and improve the enthusiasm of villagers' participation.

### 2.2. Theoretical Correlation Framework

In today's context of urbanization, traditional villages are characterized by locality, self-organization, vulnerability, and imbalance. Resilience theory aligns to some extent with practical needs for the protection and development of traditional villages and in solving rural problems. The present study was conducted in an attempt to build a correlation framework of "resilience theory-traditional village protection and development" from the dual aspects of "element correlation" and "role correlation" (Figure 2).

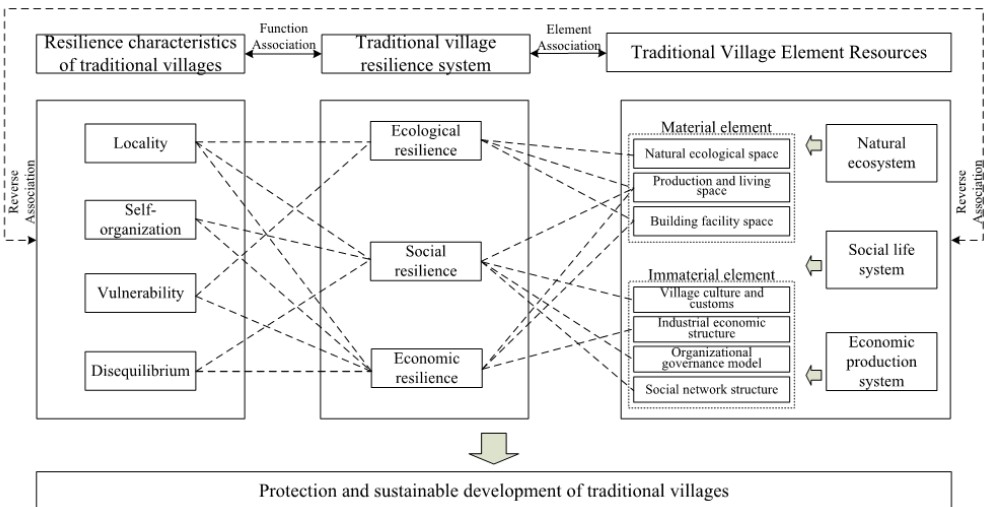

**Figure 2.** Correlation framework of resilience theory–protection and development of traditional villages.

### 2.2.1. Element Association

There are two major inputs necessary for the protection and development of traditional villages: material elements and non-material elements. Material elements, including natural ecological space, production and living space, and building facility space, are the dominant representation elements in the village which most reflect its style, pattern, and format. Non-material elements, including village culture and customs, social network structure, industrial economic structure, social organization, and governance are the hidden driving factors affecting sustainable development. Here, material elements and non-material elements are coupled with ecological resilience, social resilience, and economic resilience to build a correlation structure between the traditional village resilience system and elemental resources.

### 2.2.2. Effect Correlation

Interactions between the resilience characteristics of traditional villages (locality, self-organization, vulnerability, and imbalance) and various elements of the village system create a buffer against factors of risk and instability. Thus, resilience allows for certain resistance, adaptation, transformation and renewal capabilities that may bolster the sustainable protection and development of traditional villages. As shown in Figure 2, there is a two-way interaction between rural resilience and traditional villages in terms of "element association" and "role association". Simultaneously, the organizational modes of material and non-material elements reversely affect resilience, thus promoting a dynamic balance in the traditional village system.

## 3. Study Area

Lishui City was selected as the study area (Figure 3). Lishui City is located in the southwest of Zhejiang Province with coordinates between 118°41′–120°26′ E and 27°25′–28°57′ N. In terms of administrative division, Lishui City has one municipal district, Liandu, and seven counties: Qingtian County, Jinyun County, Suichang County, Songyang County, Yunhe County, Qingyuan County, and Jingning County. Longquan City, with a municipal area of 17,300 square kilometers, is the area's sole county-level city. Lishui is located at the junction of Zhejiang Province and Zhejiang–Fujian Province, bordering Wenzhou City in the southeast, Ningde City and Nanping City in Fujian Province in the southwest, Quzhou City in the northwest, Jinhua City in the north, and Taizhou City in the northeast. Landform-wise, the Lishui mountain range belongs to the Wuyi Mountain system, which runs from southwest to northeast and extends outward to the northwest, southwest, and northeast. It is a typical representative of southwest mountain areas in Zhejiang Province. In terms

of its hydrological climate, Lishui City belongs to the subtropical monsoon climate zone. As the terrain is dominated by mountains and hills, it has significant three-dimensional mountain climate characteristics.

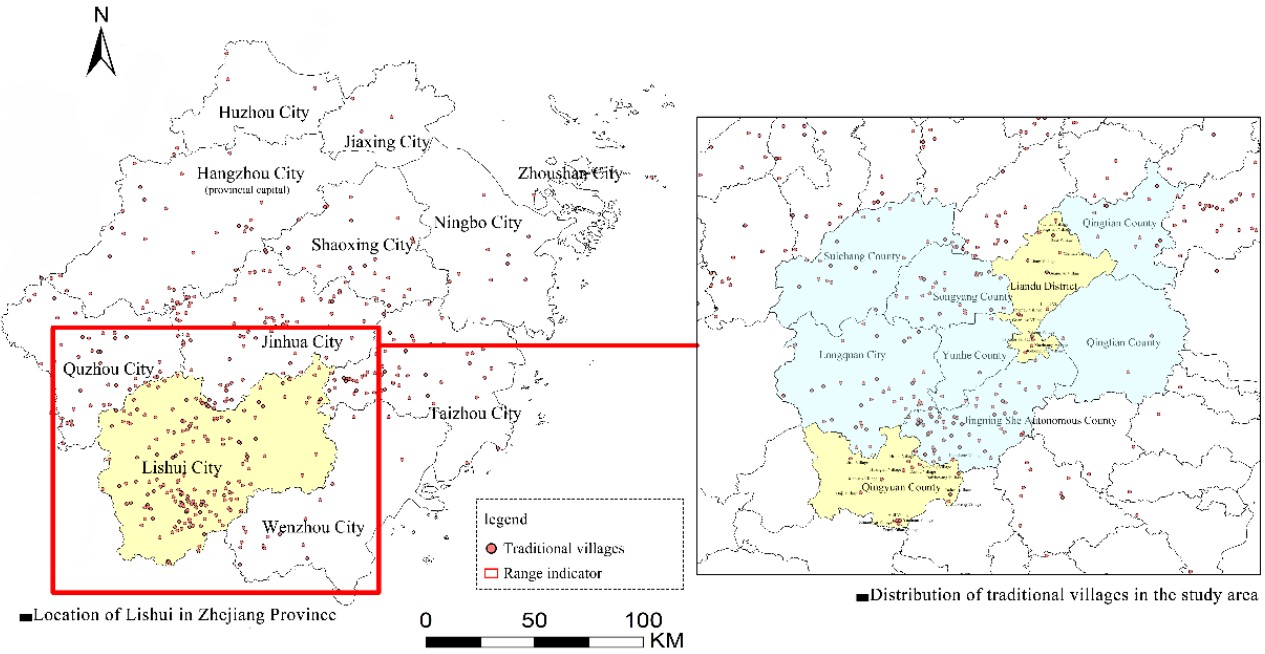

**Figure 3.** Location of study area.

Fuzzy-set QCA (fsQCA) requires that selected cases have sufficient similarity (in background or characteristics) to be comparable in certain dimensions. To strengthen the external effectiveness of the results and facilitate the full comparison of cases, case selection should also pursue the maximum heterogeneity between cases [22]. Twenty-eight national traditional villages in Liandu District and Qingyuan County, which contains Lishui City, were selected as research cases for the purposes of this study accordingly (Figure 3). Among them, Liandu District, as the administrative center of Lishui City, intuitively reflects the impact of urbanization on traditional suburban villages. Qingyuan County is located to the east of Lishui City and at the junction of Zhejiang and Fujian provinces. Most of its traditional villages are located in regions with relatively remote geographical locations and ineffective traffic and economic conditions. Therefore, it is highly representative of the characteristics at play in the protection and development of traditional villages.

## 4. Research Methods and Data Sources

### 4.1. Research Methods

QCA is a set theory configuration analysis method based on Boolean algebra. It was first proposed by Charles C. Ragin, an American sociologist, in his 1987 book Comparative Methods: Beyond Qualitative and Quantitative Strategies; it is regarded as a standard method integrating quantitative and qualitative analysis [23]. At its core, in QCA, the generation of a result may be affected by multiple cause combinations and paths. It aims to explore the collective relationship between cause variables and configuration/result variables from small-sample data through the integration of empirical data and relevant theories, thus revealing multiple concurrent causal relationships and causal asymmetry [24–26]. In a traditional research process, conversely, researchers typically regard each antecedent condition as an independent cause of the result; that is, each cause can affect the production of the result and a simple causal relationship is assumed, thus ignoring the fact of limited diversity [27].

As a type of QCA, fsQCA mainly uses the concept of membership degree to assign values between cause and result variables (0–1). By combining the theoretical advantages

of case study, quantitative research, and other methods, the research object of this work was analyzed from a holistic perspective to clearly reveal diversity and causal complexity among the results [28]. Due to the complexity and diversity of the factors affecting the protection and development of traditional villages, the fsQCA method is suitable for this research.

### 4.2. Data Source

The data used in this study include literature data, local statistical data, interview information data, and image information data. The literature review centered on the collation and collection of relevant data (e.g., from books, journal articles, dissertations) surrounding resilience theory, traditional village protection and development, QCA methods, and other relevant theoretical methods. Next, the data statistics yearbook, archives, news reports, policy documents, and other information regarding Lishui City were gathered to extract information related to each research case while ensuring the objectivity and authenticity of descriptive statistics for each case. Field research and data collection across 28 traditional villages in Liandu District and Qingyuan County were then conducted for approximately three months from June to September 2021. Semi-open interviews [29–32] were also conducted with >150 administrative staff members in each village to build a basic overview of each village, its current problems and contradictions, and future development needs in detail (See Appendix A Table A1 for the content of the interviews). Drones, cameras, and other equipment were used to collect images reflecting current situations, historical buildings, environmental patterns, and other photographic information for each village, then sorted to form image data sets further supporting this work.

## 5. Research Design

### 5.1. Variable Selection

There are five theoretical methods for variable selection in QCA:

(1) The problem-oriented method, i.e., mining relevant conditional variables in research problems;
(2) The research framework method, which determines antecedent conditions based on an existing research framework;
(3) The theoretical perspective method, where for specific research problems the same or different theoretical perspectives are used to form relevant antecedents;
(4) Literature induction, which summarizes important conditions from existing relevant literature or a literature review;
(5) The phenomenon summary method, which involves forming or obtaining credible conditions from research phenomena [22].

In this work, we utilized all of the above methods with reference to the literature to select variable conditions. We reached five reason variables (Figure 4) from three levels of ecological resilience, social resilience, and economic resilience according to the correspondence between the number of research samples and the number of variables [33,34] (Table 2) in the QCA of fuzzy sets. These mainly included: (1) natural disaster response (simplified as "ZRZH"); (2) landscape (environmental) protection (JGHJ); (3) social organization governance (SHZZ); (4) cultural element inheritance (WHYS); and (5) village industry development (CLCY). The outcome variable was the level of protection and development of traditional villages (BHFZ).

### 5.2. Complex Causal Model for Protection and Development of Traditional Villages

Variable selection is the basis for the use of the fsQCA method. In this work, we used the above five cause variables and one outcome variable to jointly construct a complex causal model (Figure 5). From the perspective of resilience theory, we analyzed the factors that affect village protection and development. We sought to determine how to improve the capacity of natural resource integration and infrastructure construction via ecological resilience, social resilience, and economic resilience; we also sought to determine how social

organization interaction, cultural inheritance, and industrial development abilities may allow for sustainable village development.

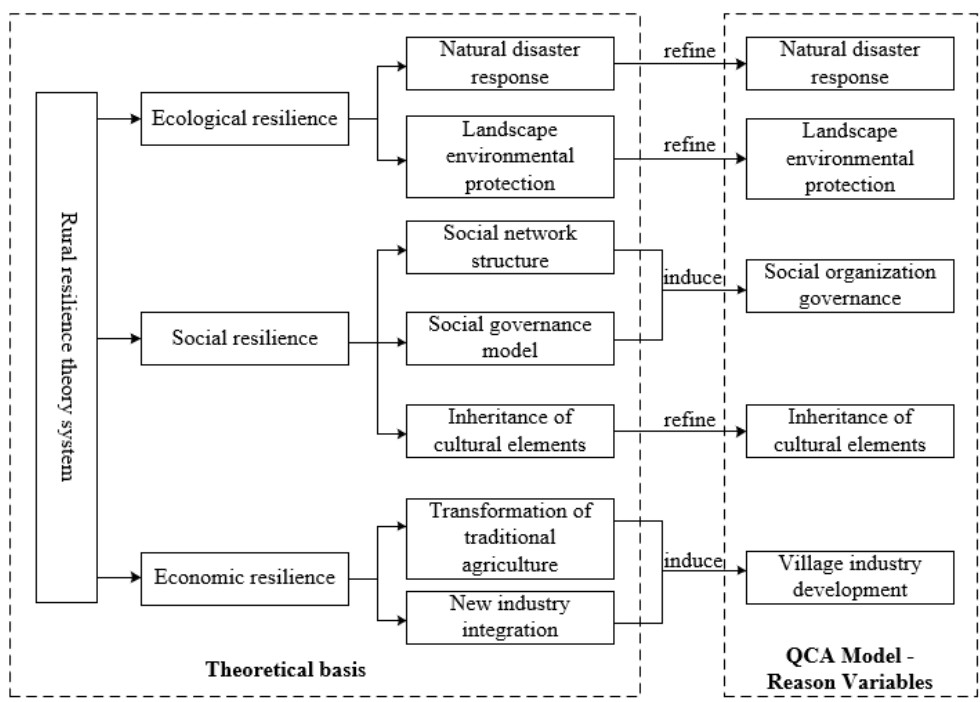

**Figure 4.** Cause variable selection.

**Table 2.** Corresponding relationship between condition quantity of variables and sample quantity for QCA of fuzzy sets.

| Variable Condition Quantity | Sample Quantity |
|---|---|
| 4 variable conditions | 10–12 samples and above |
| 5 variable conditions | 13–16 samples and above |
| 6 variable conditions | 16–25 samples and above |
| 7 variable conditions | 27–29 samples and above |
| 8 variable conditions | 36–45 samples and above |

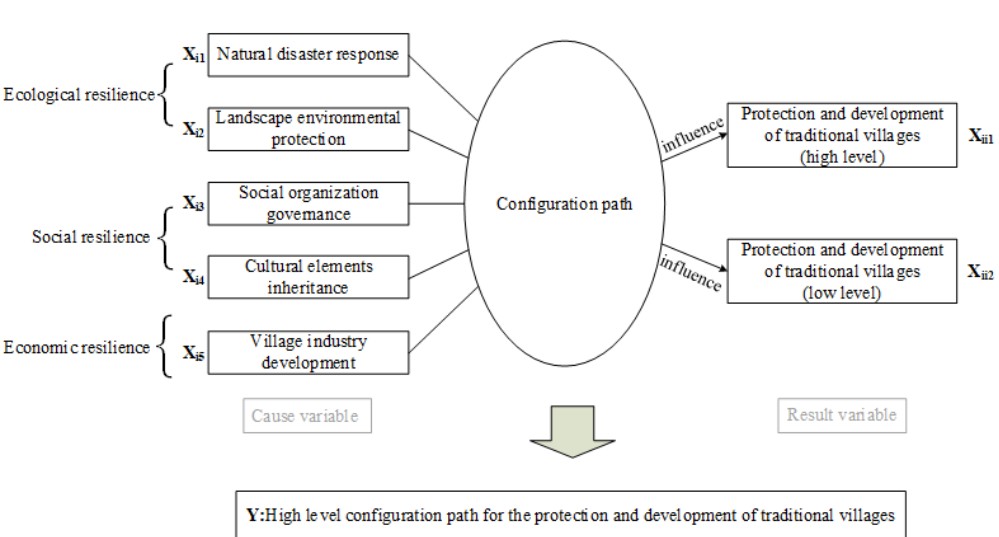

**Figure 5.** Complex causal model of traditional village protection and development.

### 5.3. Variable Calibration

Variable calibration is an important step in fsQCA. It is mainly based on algorithmic rules under which the target is transformed into a result set. The descriptive statistics and data of the case are integrated and optimized. The variable is calibrated by setting anchor points, then converted into a score representing the membership of the set. Generally, the value is between (0–1). The unique score value of each case represents its membership degree to the corresponding result. The qualitative context description gives two main types of relationship: full membership and non-membership. As shown in Table 3, if the membership score of a case is (0), then the case does not belong to the result set at all. If the membership value is [1], it means that this case entirely belongs to this set. Generally, (0.5) is the largest fuzzy point between the two, also known as the intersection. The QCA of fuzzy sets can clearly and intuitively show the membership relationship of different cases to the same antecedent condition. Consistency and coverage can be checked in the fsQCA3.0 software (The software was developed by the American scholar Charles Ragin et al.) algorithm.

There are three main states of traditional village protection and development under analysis here: high level, low level, and fuzzy level [35,36]. Thus, the ternary fuzzy set has a certain amount of applicability for variable assignment in this work (Table 3) [37]. However, there are some problems in the villages we analyzed such as incomplete data collection and lagging data statistics. Many variables could not be measured with objective data. In the process of variable assignment, we followed two principles: (1) if there were objective and authoritative data that could directly or indirectly measure variables, we assign values according to these data; and (2) variables that could not be measured by objective data were assigned by "factual data (descriptive information)" to make them "relatively objective under subjective conditions". According to our 28 research cases and pieces of information we gathered via interviews, the variables were calculated and assigned with reference to the Chinese Traditional Village Evaluation and Identification Indicator System (Trial) and relevant literature. The textual information was converted into data and input to the variable calibration table shown in Table 4.

**Table 3.** Classification of fuzzy sets (data source: reference [37]).

| Three-Valued Fuzzy Sets | Four-Valued Fuzzy Sets | Six-Valued Fuzzy Sets | "Continuous" Fuzzy Sets |
|---|---|---|---|
| 1 = fully subordinate | 1 = fully subordinate | 1 = fully subordinate | 1 = fully subordinate |
| 0.5 = neither complete nor complete subordination | 0.67 = partial subordination | 0.9 = non-subordinate | Partial subordination: 0.5 < Xi < 1 |
| | 0.33 = not affiliated | 0.6 = some subordination | 0.5 = intersection: neither subordination nor non-subordination |
| 0 = not affiliated at all | 0 = not affiliated at all | 0.4 = some are not affiliated | Partial non-subordination: 0 < Xi < 0.5 |
| | | 0.1 = very non-subordinate 0 = not affiliated at all | 0 = not affiliated at all |

### 5.4. Truth Table

The truth table is a configuration set of antecedents related to sample results which clearly shows the sample cases distributed on all configurations [38]. In this work, we analyzed the impact of five cause variables on the protection and development of traditional villages in fsQCA3.0 software. Based on the suggestions of Fiss, Du Yunzhou, and other scholars [33,39], we set the consistency threshold and PRI consistency threshold to 0.8 and 0.75, respectively. We also set the case threshold to 1. Finally, we obtained a truth table with 25 lines representing all configurations of the antecedent conditions after removing the logical remainder (Table 5).

**Table 4.** Variable calibration.

| Case Number | Cause Variable | | | | | Result Variable |
|---|---|---|---|---|---|---|
| | **ZRZH** | **JGHJ** | **SHZZ** | **WHYS** | **CLCY** | **BHFZ** |
| 01 | 1 | 1 | 0.499 | 1 | 0.499 | 0.93 |
| 02 | 0.499 | 1 | 1 | 0.499 | 0.499 | 0.83 |
| 03 | 1 | 1 | 1 | 1 | 1 | 0.95 |
| 04 | 1 | 0.499 | 0.499 | 0 | 0 | 0.09 |
| 05 | 0.499 | 1 | 0.499 | 1 | 0.499 | 0.83 |
| 06 | 0 | 0.499 | 0 | 0 | 0 | 0.95 |
| 07 | 0.499 | 0.499 | 1 | 0.499 | 1 | 0.09 |
| 08 | 0.499 | 0.499 | 0 | 0 | 0.499 | 0.56 |
| 09 | 0.499 | 0.499 | 0.499 | 1 | 1 | 0.05 |
| 10 | 0.499 | 1 | 0.499 | 0.499 | 1 | 0.46 |
| 11 | 1 | 1 | 1 | 0.499 | 0 | 0.1 |
| 12 | 1 | 0.499 | 0.499 | 1 | 1 | 0.33 |
| 13 | 0.499 | 1 | 0.499 | 0 | 0 | 0.42 |
| 14 | 1 | 1 | 1 | 1 | 1 | 0.54 |
| 15 | 1 | 1 | 0.499 | 0.499 | 0.499 | 0.67 |
| 16 | 0.499 | 1 | 0.499 | 1 | 0.499 | 0.09 |
| 17 | 1 | 0.499 | 1 | 1 | 1 | 0.63 |
| 18 | 1 | 1 | 1 | 0.499 | 0.499 | 0.55 |
| 19 | 0.499 | 1 | 0.499 | 1 | 1 | 0.77 |
| 20 | 0 | 1 | 0.499 | 0.499 | 0 | 0.57 |
| 21 | 0.499 | 0.499 | 0 | 0 | 0 | 0.09 |
| 22 | 1 | 1 | 1 | 0.499 | 1 | 0.61 |
| 23 | 0.499 | 1 | 1 | 1 | 1 | 0.77 |
| 24 | 1 | 1 | 1 | 1 | 0.499 | 0.48 |
| 25 | 0.499 | 1 | 0.499 | 0 | 0 | 0.08 |
| 26 | 0.499 | 1 | 0.499 | 0 | 0 | 0.09 |
| 27 | 0.499 | 1 | 1 | 0 | 0.499 | 0.7 |
| 28 | 0.499 | 1 | 1 | 0.499 | 0 | 0.05 |

**Table 5.** Truth table.

| ZRZH | JGHJ | SHZZ | WHYS | CLCY | Number of Cases | BHFZ | RAW [1] | PRI [2] | SYM [3] |
|---|---|---|---|---|---|---|---|---|---|
| 1 | 0 | 0 | 1 | 1 | 1 | 1 | 1 | 1 | 1 |
| 1 | 0 | 1 | 1 | 1 | 1 | 1 | 1 | 1 | 1 |
| 0 | 1 | 0 | 1 | 1 | 1 | 1 | 1 | 1 | 1 |
| 0 | 1 | 1 | 1 | 1 | 1 | 1 | 1 | 1 | 1 |
| 0 | 0 | 1 | 0 | 1 | 1 | 1 | 0.998 | 0 | |
| 0 | 1 | 0 | 1 | 0 | 2 | 1 | 0.997 | 0 | 0 |
| 0 | 0 | 0 | 1 | 1 | 1 | 1 | 0.996 | 0 | 0 |
| 1 | 1 | 1 | 0 | 1 | 1 | 1 | 0.857 | 0.750 | 0.750 |
| 1 | 1 | 1 | 1 | 0 | 1 | 0 | 0.777 | 0.499 | 0.499 |
| 1 | 1 | 0 | 1 | 0 | 1 | 0 | 0.750 | 0.501 | 0.501 |
| 0 | 1 | 0 | 0 | 1 | 1 | 0 | 0.499 | 0 | 0 |
| 0 | 1 | 1 | 0 | 0 | 3 | 0 | 0.428 | 0.334 | 0.334 |
| 1 | 1 | 1 | 0 | 0 | 2 | 0 | 0.399 | 0.249 | 0.249 |
| 0 | 1 | 0 | 0 | 0 | 4 | 0 | 0.142 | 0.030 | 0 |
| 0 | 0 | 0 | 0 | 0 | 3 | 0 | 0 | 0 | 0 |
| 1 | 0 | 0 | 0 | 0 | 1 | 0 | 0 | 0 | 0 |
| 1 | 1 | 0 | 0 | 0 | 1 | 0 | 0 | 0 | 0 |

[1] The degree of membership in the space angle of the raw consistency pointer is the degree of consistency of the membership subset of the result. [2] PRI consistency refers to another consistency calculation method based on fuzzy sets with reduced quasi proportion in error calculation. [3] SYM consistency refers to the consistency substitution measure of fuzzy sets based on the symmetrical version of PRI consistency.

## 6. Results

### 6.1. Necessity Analysis

Necessity analysis is a key step in the QCA of fuzzy sets, as it determines whether the cause variable constitutes a necessary condition for the result variable (Figure 6). Generally, consistency is used for judgment [30–32]. If the consistency index is greater than 0.9, the variable is considered a necessary condition and must exist when the result exists [23,40]. Consistency is calculated as follows:

$$\text{Consistency}(X_i \leq Y_i) = \frac{\sum[\min(X_i, Y_i)]}{\sum X_i} \tag{1}$$

where Xi and Yi correspond to the respective subordinate scores and the value range of Consistency is (0–1).

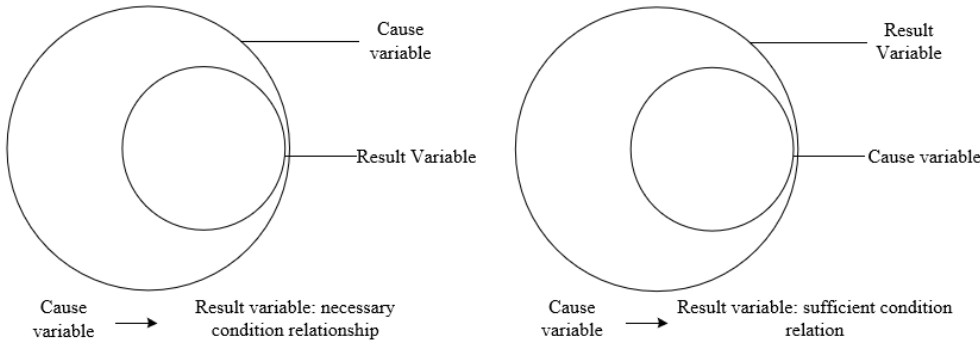

**Figure 6.** Relationship between cause and result variables.

After completing the analysis of necessary conditions, the extent to which the cause variable or combination of variables can explain the result variable further depends on the coverage [41–43]. Higher coverage indicates stronger explanatory power in the result variable. Coverage is calculated as follows:

$$\text{Coverage}(X_i \leq Y_i) = \frac{\sum[\min(X_i, Y_i)]}{\sum Y_i} \tag{2}$$

where Xi and Yi correspond to the respective subordinate scores and the value range of Coverage is (0–1).

We used fsQCA3.0 to calculate the consistency and coverage of each cause variable. As shown in Table 6, with the result variable "high-level protection and development", only the consistency of the single variable of landscape protection (JGHJ) exceeded 0.9 (i.e., 0.930789), which theoretically indicates that this variable constitutes a necessary condition for the protection and development of traditional villages. The protection of landscapes in traditional villages thus appears to significantly influence their overall protection and development. The consistency of other variables was less than 0.9 but more than 0.7, indicating that these single cause variables do play a certain role in promoting the results, but do not sufficiently explain the emergence of the result variables [41]. Instead, different cause variables work together to produce results.

We further analyzed the impact and characteristics of different variable configurations on the protection and development of traditional villages. As shown in Table 7, when the result variable was set to "~BHFZ", the consistency of no cause variable exceeded 0.9, indicating that the necessary conditions do not exist. However, the consistency of "low-level village industry development" reached 0.814924, i.e., was close to 0.9, which shows that low-level village industry development has a significant restrictive effect on the protection and development of traditional villages. In addition, none of the variables has an obvious restrictive effect on the protection and development of traditional villages.

**Table 6.** Necessary conditions for each cause variable (BHFZ, high-level protection and development).

| Variable | Consistency | Coverage |
|---|---|---|
| ZRZH | 0.827215 | 0.648526 |
| ~ZRZH | 0.344949 | 0.525381 |
| JGHJ | 0.930789 | 0.562229 |
| ~JGHJ | 0.138076 | 0.499251 |
| SHZZ | 0.896288 | 0.702602 |
| ~SHZZ | 0.275876 | 0.420267 |
| WHYS | 0.861786 | 0.806210 |
| ~WHYS | 0.310378 | 0.359581 |
| CLCY | 0.827422 | 0.827479 |
| ~CLCY | 0.310309 | 0.332889 |

**Table 7.** Necessary conditions for each causal variable (~BHFZ, low-level protection and development).

| Variable | Consistency | Coverage |
|---|---|---|
| ZRZH | 0.665680 | 0.486448 |
| ~ZRZH | 0.519026 | 0.736836 |
| JGHJ | 0.851421 | 0.479368 |
| ~JGHJ | 0.222461 | 0.749750 |
| SHZZ | 0.591723 | 0.432358 |
| ~SHZZ | 0.592982 | 0.842006 |
| WHYS | 0.406944 | 0.354851 |
| ~WHYS | 0.777761 | 0.839875 |
| CLCY | 0.332840 | 0.310262 |
| ~CLCY | 0.814924 | 0.814864 |

*6.2. Calculation Results*

　　Based on the fsQCAmodel {Model: BHFZ = f (ZRZH, JGHJ, SHZZ, WHYS, CLCY)} of traditional village protection and development, and after analysis in fsQCA3.0 software, we obtained the result sets shown in Tables 8–10. The counterfactual case (i.e., the logical remainder) contains more or less content. The complex solution contains all configuration paths without a logical remainder. The simplified solution is allowed to include all logical remainders, but does not consider their justification (i.e., existing practical knowledge). The intermediate solution only contains the logical remainder that meets the expectation of the theoretical direction [44]. As shown in Table 6, since "landscape protection" is a necessary condition for the high-level protection and development of traditional villages, it was necessary to make directional choices when outputting intermediate solutions. However, there was no clear theoretical expectation between the other four variables and high-level results, so we made no specific counterfactual analysis in operating the software (Table 11).

**Table 8.** Complex solution.

| Conditional Configuration | Raw Coverage | Unique Coverage | Consistency |
|---|---|---|---|
| ~ZRZH*JGHJ*~SHZZ*WHYS | 0.206735 | 0.034433 | 0.998667 |
| ~JGHJ*~SHZZ*WHYS*CLCY | 0.069004 | 0 | 0.998004 |
| ZRZH*~JGHJ*WHYS*CLCY | 0.138007 | 0.000138 | 1 |
| ZRZH*JGHJ*SHZZ*CLCY | 0.689070 | 0.069004 | 0.952408 |
| ~ZRZH*~JGHJ*SHZZ*~WHYS*CLCY | 0.034502 | 0.069022 | 0.998004 |
| ~ZRZH*JGHJ*WHYS*CLCY | 0.275738 | 0 | 1 |
| JGHJ*SHZZ*WHYS*CLCY | 0.654637 | 0.034433 | 0.95003 |
| Overall coverage: 0.758556 | | | |
| Overall consistency: 0.955996 | | | |

Note: "*" is the connector between variables and also represents the "intersection" between variables. "~" represents "non set", which means it does not exist.

**Table 9.** Intermediate solution.

| Conditional Configuration | Raw Coverage | Unique Coverage | Consistency |
|---|---|---|---|
| ~SHZZ*WHYS*CLCY | 0.241306 | 0 | 0.874687 |
| JGHJ*SHZZ*CLCY | 0.723641 | 0.069142 | 0.954579 |
| ZRZH*WHYS*CLCY | 0.689208 | 0.069004 | 0.952417 |
| ~ZRZH*JGHJ*~SHZZ*WHYS | 0.206735 | 0.034433 | 0.998667 |
| ~ZRZH*SHZZ*~WHYS*CLCY | 0.1378 | 0.069022 | 0.999499 |
| Overall coverage: 0.82742 | | | |
| Overall consistency: 0.95951 | | | |

Note: "*" is the connector between variables and also represents the "intersection" between variables. "~" represents "non set", which means it does not exist.

**Table 10.** Parsimonious solution.

| Conditional Configuration | Raw Coverage | Unique Coverage | Consistency |
|---|---|---|---|
| ~ZRZH*WHYS | 0.310171 | 0.034571 | 0.901 |
| ZRZH*CLCY | 0.758211 | 0.034433 | 0.916736 |
| SHZZ*CLCY | 0.758280 | 0.034433 | 0.916667 |
| Overall coverage: 0.827422 | | | |
| Overall consistency: 0.856867 | | | |

Note: "*" is the connector between variables and also represents the "intersection" between variables. "~" represents "non set", which means it does not exist.

**Table 11.** Directionality preset of intermediate solution.

| Cause and Effect Conditions | Leads to BHFZ Because: | | |
| | Existent | Non-Existent | Existent or Non-Existent |
|---|---|---|---|
| ZRZH | | | √ |
| JGHJ | √ | | |
| SHZZ | | | √ |
| WHYS | | | √ |
| CLCY | | | √ |

Note: Evolve fsQCA3.0 software operation process; "√" means the option is selected.

In the table, "raw coverage" refers to how many cases can be explained by a single configuration path, as the same results may exist and be repeatedly combined for interpretation. "Unique coverage" indicates how many cases can be explained through the configuration path, where a larger value leads to more case results. The overall coverage and overall consistency, respectively, represent the interpretability and credibility of the case as a whole. An overall consistency value must generally be greater than 0.8 [42].

### 6.3. High-Level Configuration Path Analysis

Our focus here is on the positive results relevant to traditional village protection and development issues, so we will center this discussion on the high-level configuration path. Core edge conditions must be involved in configuration analysis. Core conditions and edge conditions exert a combined impact on results; that is, a result can reach the same state through different initial conditions and different paths [37]. The core (or marginal) condition concept mainly originates from simple and intermediate solutions. The core condition refers to the cause condition, which has a strong causal relationship with the result and is a part of the simple/intermediate solution. The marginal condition refers to the cause condition with weak causality with the result, which can be eliminated in the simplified solution and only appears in the intermediate solution [43]. Because the intermediate solution contains some meaningful logical remainders, after comprehensive consideration, taking the intermediate solution as the main analysis object can make the results more universal.

There were five configuration paths in the intermediate solution. However, the unique coverage of one path was 0, which means that it did not have unique explanatory power—we did not analyze this path. A total of four paths with strong explanatory power were obtained. We translate the effective path of the intermediate solution into the full Chinese name as follows: [landscape environmental protection * social organization governance * village industry development (path 1) + natural disaster response * cultural element inheritance * village industry development (path 2) + ~natural disaster response * landscape environmental protection *~social organization governance * cultural element inheritance (path 3) + ~natural disaster response * social organization governance * ~cultural element inheritance * village industry development (path 4)] → [Protection and development of traditional villages]. The simplified solution translates to: [~natural disaster response * cultural element inheritance (path 1) + natural disaster response * village industry development (path 2) + social organization governance * village industry development (path 3)] → [traditional village protection and development].

From the perspective of resilience theory, our five reason variables belong to "ecological resilience, social resilience, and industrial resilience". The five variables are imbalanced; that is, some are effectively "foundational", more advantageous, and easier to develop. Some aspects show notable weaknesses and are slower to develop than others. According to the configuration of the intermediate and reduced solutions and the symbolic representation of the solution table introduced by Ragin and Fiss, we grouped solutions according to their core conditions. A black circle ("●") here indicates that there is a condition, the circle with a cross ("⊗") indicates that there is no condition, a large circle indicates the core condition, and a small circle indicates an edge condition. Variables that can exist or not exist are expressed as blanks [45]. The notes under Table 12 show the core and edge conditions in the configuration of influencing factors for traditional village protection and development, including four configuration paths that explain the generation of high-level traditional village protection and development.

**Table 12.** High-level configuration path analysis.

| Variable | High-Level Configuration Path | | | |
|---|---|---|---|---|
| | 1 | 2 | 3 | 4 |
| Natural disaster response | | ● | ⊗ | ⊗ |
| Landscape protection | ● | | ● | |
| Social organization governance | ● | | ⊗ | ● |
| Cultural element inheritance | | ● | ● | ⊗ |
| Village industry development | ● | ● | | ● |
| **Consistency** | 0.954579 | 0.952417 | 0.998667 | 0.999499 |
| **Original coverage** | 0.723641 | 0.689208 | 0.206735 | 0.1378 |
| **Unique coverage** | 0.069142 | 0.069004 | 0.034433 | 0.069022 |
| **Overall consistency** | | 0.95951 | | |
| **Overall coverage** | | 0.82742 | | |

As shown in Table 12, from the perspective of resilience theory, traditional villages achieve high-level protection and development mainly through four configuration paths, which can cover 82.7442% of cases as a sufficient condition for high-level results. We further analyzed these four paths as discussed below.

**Path 1: [Landscape Environmental Protection * Social Organization Governance * Village Industry Development].**

On this path, if a village can protect its landscape from being damaged, the social organization is effectively managed, the social network structure is relatively balanced, and there are certain industrial development advantages and foundations, high-level protection and development can be promoted. The consistency of this path was 0.954579 (>0.8), which is a strong consistency standard. The three variables of landscape protection, social organization governance, and industry development all appeared in the configuration path

with core conditions. Therefore, the path was mainly dominated by landscape protection, social organization governance, and industry development.

Historical buildings in the study area were mostly brick and wood structures or rammed earth structures, which face significant challenges from mold, insect infestation, and corrosion. Village residents also tended to have lower education levels and advanced age; their awareness of village protection tended to be fairly weak. In efforts to improve their living conditions despite their limited resources, some villagers choose to demolish old structures and rebuild new ones or "self-repair" their homesteads. This appears, to some extent, to irreversibly damage the overall texture of the traditional village thus affecting the ability to protect its landscape.

This path also emphasizes the importance of villagers and other subjects involved in the protection and development of traditional villages. Social organization governance and policy support can slow down the trend of population loss in villages. Development of the village's industry also may promote sustainable development. Only when the village has the necessary industrial/economic foundation can capital be effectively injected into its protection, repair, and construction, thereby improving its overall sustainable development rather than providing purely static protection.

The coverage rate of this path ranked first among the four paths, indicating that this configuration is best suited to the protection and development of traditional villages among the four paths we analyzed.

**Path 2: [Natural disaster response * Cultural element inheritance * Village industry development].**

The consistency value of this path was 0.952417 (>0.8), which indicates good consistency. Natural disaster response and village industry development are the core conditions of this path while cultural element inheritance is a marginal condition. This path contains a combination of core conditions and marginal conditions. A village on this path can cope with and prevent natural disasters while maintaining a certain amount of industry development. If the cultural elements of the village are also strong, these elements together will promote its protection and development.

As most traditional villages in the study area were located in mountainous areas, their infrastructures and public service facilities were relatively ineffective. Different types of natural disasters exert a significant impact on traditional villages, threatening the lives and property of residents and the overall environmental pattern of the village. It is crucial to strengthen a village's natural disaster response in order to effectively protect and develop it. The inheritance of cultural elements includes both material and intangible objects. It aims to improve the humanistic status and value of the village while allowing its culture to continually evolve. Protecting the village from natural disasters allows it to preserve its culture effectively.

**Path 3: [~Natural disaster response * Landscape environmental protection *~Social organization governance * Cultural element inheritance].**

The consistency value of this path was 0.998667 (>0.8), which also indicates good consistency. This path highlights the synergy of the two elements of "landscape protection" and "cultural element inheritance". A traditional village on this path can also be protected and developed at a high level if it focuses on the inheritance of cultural elements and the complete preservation of the landscape. However, on this path, the variable condition of "natural disaster response" appears as a "core condition that does not exist"; further, this path can only effectively explain 20.6735% of the sample cases, indicating that most do not follow it to achieve the protection and development of traditional villages but are specific villages that are not threatened by natural disasters. This path appears similar to "rescue" protection and development measures. More emphasis is placed on the protection of their original ecological environment, architectural space, and pattern from constructive and man-made damage in ensuring their continued authenticity. Secondly, the villagers' cultural identity and sense of belonging can be enhanced by focusing on the cultural elements of the village.

**Path 4: [~Natural disaster response * Social organization governance *~Cultural element inheritance * Village industry development].**

This path is characterized by two resilience elements; namely, "social organization governance" and "village industry development" in the construction of "social resilience" and "economic resilience". The consistency value of this path was 0.999499 (>0.8), which indicates good consistency. However, the coverage of this path is relatively low, as it can only explain 13.78% of the cases (i.e., 3–4 cases), so it was only partially applicable to the villages under analysis here. This path appears to center more on the development of traditional villages than their protection and is applicable to villages with serious population loss and single social structures. Villages on this path must self-develop industries, attract government support, and stabilize their own social systems in order to continue developing.

The first and second paths, as described above, had the highest coverage and could explain 72.3641% and 68.9208% of the cases, respectively. Therefore, they are more universal and have greater value for the protection and development of traditional villages. The five elements of natural disaster response, landscape protection, social organization governance, cultural element inheritance, and industry development appeared in combinations of core or marginal conditions in different paths and played different roles on different paths. The reason variable of "village industry development", as an important indicator in the resilience of the traditional village economy, appeared in multiple paths in the state of "existence" with zero instances of "no integration". To this effect, a village requires a strong industrial sector as the basis for sustainable development and enhanced internal power. To stimulate the inherent vitality of traditional villages via industry will promote their survival and continued development in the context of urbanization.

## 7. Discussion

### 7.1. Traditional Village Conservation and Development Strategy

According to the results of the above study, the reason variable "landscape environmental protection" emphasizes that the ecological background and the architectural pattern are the important basic conditions for conservation and development. As the main component of the high-level histological path, "village industrial development" emphasizes that traditional villages should actively explore their own industrial characteristics and industrial resources, so that they can become a realistic guarantee for the benign growth and sustainable development of traditional villages. Factors such as natural disaster response, social organization governance, and cultural heritage are distributed in each path, which indicates that the factors influencing the protection and development of traditional villages are rich and complex. The final framework of traditional village conservation and development from the perspective of resilience theory is "based on landscape environmental protection, guaranteed by village industry development, and other influencing factors (natural disaster response, social organization, and cultural heritage)" (Figure 7).

Therefore, for the convenience of the study, the four high-level grouping paths were divided into four types of traditional village conservation and development models, and each path named according to the initials of its core conditions (see Table 13 and Figures 8–11 for details). At the same time, this paper analyzes the villages and village characteristics applicable to each model from the perspective of resilience theory, and specifically proposes conservation and development strategies for different village types (Tables 14 and 15).

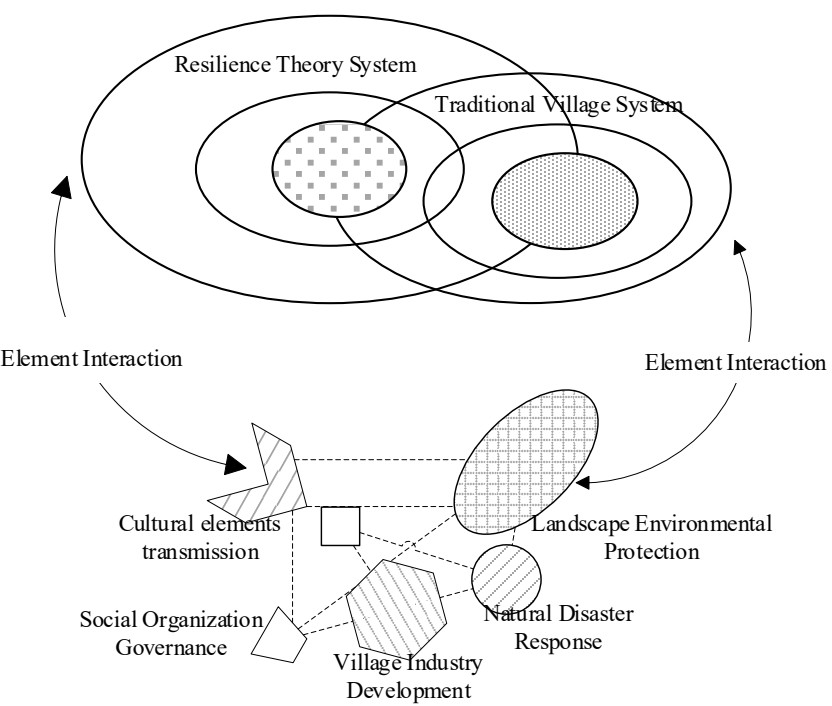

**Figure 7.** Protection and development framework of traditional villages from the perspective of resilience theory.

**Table 13.** Protection and development model of traditional villages.

| Configuration Path | Mode | Applicable Villages (Serial Number) |
|---|---|---|
| Path 1: [Landscape Environmental Protection * Social Organization Governance * Village Industry Development] | JSC Model | Xixi Village (01), Guanling Village (04), Kuchuan Village (05), Gongshan Village (06), Saikeng Village (08), Meitian Village (12), Hengkeng Village (19), Huangtan Village (21), Pao Tau Village (22) Ruo Hang Village (24), Turning Water Village (25), Yuchuan Village (26) |
| Path 2: [Natural disaster response * Cultural element inheritance * Village industry development] | Z&C Mode | Weantou village (03), Lianji Village (10), Liang Village (11), Daji Village (14), Xichuan Village (16), Nanyang Village (17), Houxi Village (18), Plain Village (28). |
| Path 3: [~Natural disaster response * Landscape environmental protection *~Social organization governance * Cultural element inheritance] | J&W Mode | Guanqiao Village (02), Kukeng-yang Village (07), Xiazhuang Village (09), Bai Zheyang Village (20), Batou Village (23) |
| Path 4: [~Natural disaster response * Social organization governance *~Cultural element inheritance * Village industry development] | S&C Mode | Xikeng Village (13), Jiji Village (15), Fengjiashan Village (27) |

Note: "*" is a conjunction between variables and indicates the "intersection" relationship between variables.

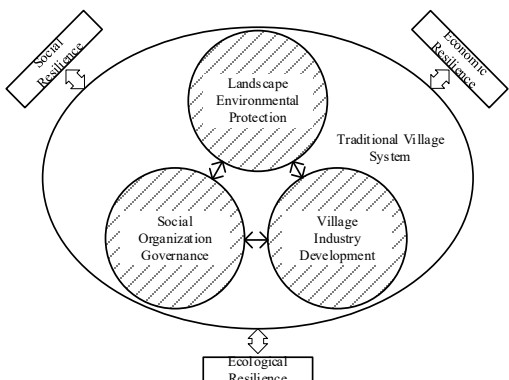

**Figure 8.** JSC mode diagram.

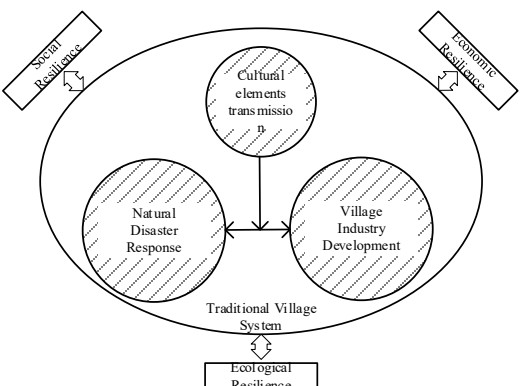

**Figure 9.** Z&C mode diagram.

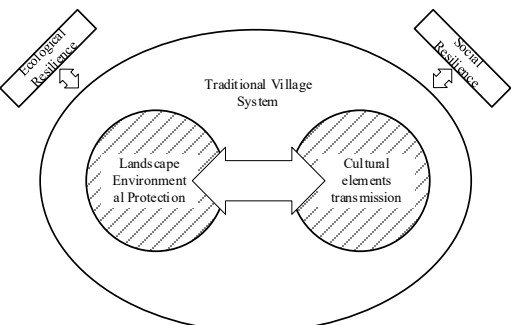

**Figure 10.** J&W mode diagram.

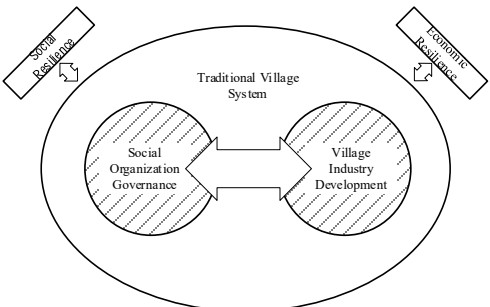

**Figure 11.** S&C mode diagram.

**Table 14.** Action mechanism of each path mode.

| Path Model | Traditional Village Conservation and Development Strategy |
|---|---|
| JSC Model | ① Construct a landscape protection framework with anti-disturbance to maintain the characteristics of traditional village style; ② attract outside capital, increase efforts to protect and repair village buildings, and provide the necessary infrastructure and public service facilities; ③ reconcile village preservation and development requirements and seek diversified village economic resilience building. |
| Z&C Mode | ① Vigorously enhance the vitality of folk culture and ensure the preservation and transmission of cultural elements in villages; ② develop village industries according to local conditions to ensure the continuation of the production and lifestyle of the indigenous people of traditional villages. |
| J&W Mode | ① Improve the ecological safety of traditional villages and enhance their buffering capacity against adverse impacts; ② create a unique village culture brand to improve the identity and visibility of the village culture; ③ remodel the characteristic space of the village and enhance the vitality of the spatial environment. |
| S&C Mode | ① Promote the transformation and development of villages according to the needs of the times and alleviate the phenomenon of village population loss; ② strengthen the construction of various types of protection systems to enhance the social resilience of villages. |

**Table 15.** Action mechanism of each path mode.

| Path Model | Ecological Dimension | Social Dimension | Economic Dimension | Applicable Village Characteristics |
|---|---|---|---|---|
| JSC Model | ↑ ↑ | ↑ | ↑ | ① Village landscape environment is vulnerable to external interference and impact; ② inadequate infrastructure and public service facilities; ③ village architecture faces huge financial pressure for protection and repair; ④ urgent need to explore local industrial resources for village development. |
| Z&C Mode | ↑ | ↑ | ↑ | ① Villages contain a large number of historical and cultural relics, including both material and immaterial culture, but there is still much room to improve the cultural vitality of villages; ② some villages blindly develop tourism industries, resulting in a series of problems such as insufficient excavation of village characteristics, lack of tourism infrastructure supply and poor tourism vitality. |
| J&W mode | ↑ ↑ | ↑ | ↑ | ① Susceptible to different types of natural disasters and interference, increasing the difficulty of traditional village protection; ② have certain cultural connotation characteristics, but lack of necessary excavation and inheritance; ③ villages are mostly protected in a static way, and the overall spatial vitality is relatively weak. |
| S&C mode | ↑ | ↑ | ↑ | ① Population loss is relatively serious; ② various social security facilities are not perfect, and the gap between urban and rural areas is significant. |

Note: The arrow indicates the meaning of "elevation".

*7.2. Future Outlook*

In the context of rapid urbanization, traditional villages face significant challenges and disturbances. China's traditional villages are currently the largest conservation space for historical agricultural civilization heritage in the world. There is serious value and significance in effectively protecting them for future generations to inherit. There may appear to be a contradiction between their protection and development, but it is possible to strike a balance so that development can become the motive for protection, while the purpose of protection can also be development. The traditions and cultural heritage of villages should be protected while their effective development is secured as a guarantee that they remain sustainable in the future.

This paper uses the fuzzy-set qualitative comparative analysis method to study the model of traditional village protection and development from the perspective of tenacity theory, which to a certain extent expands the theoretical research framework of traditional village protection and development, but there are also certain research limitations: (1) the limitation of research cases and the number of cases. Since the fuzzy-set qualitative comparative analysis method has certain limitations on the number of variables and cases, only 28 traditional villages in Liandu District and Qingyuan County of Lishui City were selected for this study, while traditional villages in other districts and counties were not covered, which to some extent affects the coverage and explanatory ability of the high-level histories; (2) there were some limitations in data acquisition. Compared with urban areas, traditional villages have problems of untimely or missing statistics, and some variables lack clear data indicators to be measured in the process of variable calibration. For this reason, this study obtained relevant information through interviews, and referred to existing literature and the QCA operation manual to set anchor points for affiliation assignment according to certain descriptive criteria, which inevitably resulted in subjectivity in the process.

For this reason, the relevant research should be further deepened in the following aspects in future research: (1) Expanding the scope of case point selection and the number of case samples, so as to improve the path of traditional village protection and development and make it more scientific, universal and operable. (2) Strengthening the construction of digital villages, collecting data and forming a database, which can not only provide strong data support for future village research, but also show the real problems intuitively, so that planning decision makers can make clearer judgments and measure choices.

**8. Conclusions**

Traditional Chinese villages face significant challenges and disturbances in the context of modern urbanization. Traditional villages have important value and significance to the nation as a whole; there is also, arguably, an inherent contradiction between their protection and their development. However, if we can balance their roles, development can also become the driving force for protection while the purpose of protection can also be development. From the perspective of resilience theory, this paper discussed the traditional village protection and development model, its influencing factors, and its complex causal mechanism. Our conclusions can be summarized as follows:

(1)   *Resilience theory is applicable to research on the protection and development of traditional villages*

In the context of modern urbanization, traditional villages are constantly impacted and disturbed by the external environment. They show characteristics of locality, self-organization, fragility, imbalance, and more, which may limit their protection and development. Resilience theory emphasizes the resistance, adaptation, and transformation abilities of a system subject to external interference, which is suitable for the research on the protection and development of traditional villages and may reveal workable information for achieving it. There is a strong correlation between resilience theory and traditional villages, so research in alignment with this concept may enhance the risk response capacity of traditional villages.

*(2)    The protection and development of traditional villages are affected by many factors, and there are many high-level configuration paths*

Under resilience theory, the five elements of natural disaster response, landscape (environmental) protection, social organization governance, cultural element inheritance, and village industry development were transformed into cause variables with the level of traditional village protection and development as an outcome variable so as to build a complex causal model of traditional village protection and development. Four high-level configuration paths were generated each with strong explanatory power. The five reason variables exist in different paths in the form of core conditions or marginal conditions; these results further illustrate that the protection and development of traditional villages are affected by multiple factors. Among them, the coverage of Path 1 and Path 2 was relatively high, which indicates that most traditional villages follow the characteristics of these two paths to a large extent in the process of protection and development from the perspective of resilience theory. Paths 3 and 4, respectively, seek optimal solutions for the protection and development of traditional villages in the absence of some reasons and conditions.

*(3)    Protection of the landscape/environment and development of the village's industry constitute the basis for effective protection and development*

As a necessary condition for the protection and development of traditional villages, "landscape protection" mainly emphasizes preservation of the ecological environment and architectural patterns of the village. In the high-level configuration, the reason variable of "industry development" emerged as the main constituent factor. To this effect, traditional villages should prioritize their own industrial sectors and industrial resources to promote their healthy growth and sustainable development.

The framework we ultimately secured for traditional village protection and development framework from the perspective of resilience theory was "based on landscape protection, guaranteed by village industry development, and focused on other influencing factors (natural disaster response, social organization governance, and cultural element inheritance)".

*(4)    Specific models and strategies for the protection and development of traditional villages are proposed based on the perspective of resilience theory*

Based on the high-level grouping path of traditional village protection and development, the protection and development models of traditional villages were specified into four models (JSC model, Z&C model, J&W model, and S&C model), and the protection and development strategies from the perspective of resilience theory were proposed according to the real problems and needs of different types of villages through typical case analysis of each model. The results show that different protection and development strategies should be adopted for different types of villages according to local conditions. For villages that develop slowly and are less influenced by the outside world, "protection before development" should be adopted to minimize the damage caused by brutal development; while for traditional villages that develop faster and are relatively more influenced by the outside environment, "protection" should be adopted for them. For the traditional villages that are developing faster and influenced by the outside environment, we should take the first-aid measures of "protection", that is, the simultaneous protection and development, to ensure that their authenticity will not be eroded.

**Author Contributions:** Conceptualization, X.W. and Q.Z.; methodology, X.W.; software, X.W.; validation, X.W. and Q.Z.; formal analysis, X.W.; investigation, X.W. and Q.Z.; resources, Q.Z.; data curation, X.W.; writing—original draft preparation, X.W.; writing—review and editing, X.W. and Q.Z.; visualization, X.W.; supervision, Q.Z.; project administration, Q.Z.; funding acquisition, Q.Z. All authors have read and agreed to the published version of the manuscript.

**Funding:** This research was funded by Scene Design and Research on Planning Method Model for Digital Driven Future Rural Scenarios; Fund Source, Basic Public Welfare Research Program of Zhejiang Province, China; Fund number, GN22E080817.

**Informed Consent Statement:** Informed consent was obtained from all subjects involved in the study.

**Data Availability Statement:** The data from the interviews and surveys are deemed private.

**Conflicts of Interest:** The authors declare no conflict of interest.

## Appendix A

**Table A1.** Interview Outline.

| Place of Interview | | |
|---|---|---|
| **Interview Subject and Background** | **Date of Interview** | **Recorder** |
| **Interview Content** | | |
| **Interview Questions** | **Answer Records** | **Remarks** |

Questions regarding natural disaster response.

(1) Are they affected by natural disasters? What is the type?
(2) Has the construction of infrastructure and public services been completed (e.g., electricity and telecommunications, water supply and drainage, road laying, etc.)?
(3) Are there enough emergency shelters and emergency measures available?
(4) Is disaster prevention and safety education regularly disseminated?

Questions about landscape protection.

(1) Are there any historical references to the spatial formation of the village? How is it preserved?
(2) Is the landscape environment in the village subject to human damage?
(3) How about the protection of historical and cultural buildings? Will the knowledge of traditional village protection be regularly promoted?
(4) What are the native and iconic tree and flower species? How is the preservation?
(5) Does the phenomenon of private construction affect the overall landscape pattern of the village?
(6) Is the necessary environmental health improvement and upgrading work carried out (e.g., road hardening, lighting, public toilets, sewage disposal, etc.)?

Questions about the governance of social organizations.

(1) What is the resident population in the village? What is the age structure?
(2) Village participation in public affairs in the village?
(3) Are there any organizations or institutions to resolve disputes between villages and maintain social order? Are there any village rules and regulations as well as policy systems?
(4) How about mutual help relationship among neighbors?
(5) Are school-age education as well as health and medical services being met?
(6) Are there outside organizations involved in village management and construction?
(7) Are there platforms and resources for young people to return to their hometowns to start their own businesses?

Questions regarding the transmission of cultural elements.

(1) What are the traditional culture and customs in the village? Are they still surviving?
(2) Is the village carrying out traditional culture transmission and protection work? Are there cultural inheritors? Are there any necessary management and protection measures?
(3) Is there any intangible cultural heritage (hand weaving, food, songs and dances, etc.)?
(4) Are cultural industries (cultural creation, photography, etc.) developed using the cultural elements of the village?
(5) How much do villagers participate in and identify with traditional culture and customs?

Questions about the industrial development of the village.

(1) What are the main industries or special industries in the village?
(2) What is the percentage of agricultural income? Is there any necessary transformation and development?
(3) How diversified are the industries in the village? Is there any integrated development according to the characteristics of the village, cultural resources, natural resources combined with tourism industry, cultural industry, etc.?
(4) Local employment opportunities?

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
