# Peer review of "Influencing Factors of Traditional Village Protection and Development from the Perspective of Resilience Theory"

_land, doi:10.3390/land11122314_

Round 1
Reviewer 1 Report
The manuscript presents a mathematical model, based on resilience theory, for identifying paths for rural village protection and development. The manuscript certainly merits publication, and the suggestions below are aimed at polishing and bringing some valuable but hidden parts so that they are clearer to the reader.
Lines 17-21: The take-home messages of the research should be clearly stated. I believe the paths should be mentioned here.
Lines 25-35. This introductory paragraph needs to be placed in lines of research, with appropriate citations.
Line 49: The acronym QCA needs to be explained here.
Lines 61-64. These affirmations need citations to support it. Please cite key references.
Line 68: Why are villages more vulnerable to uncertainties than cities?
Line 91: What kind of factors?
Line 120: Use of quotation marks is awkward. Does this refer to the model you have developed? Maybe you can mention that it is a model.
Lines 394-479: This is really interesting, but is quite hidden in the manuscript.
Lines 480-496: The discussion needs to compare the results of the present research with those of previous published research.
Lines 498-505: The conclusions section should focus on summarizing the results, not arguing on theoretical issues.
Line 514: Resilience theory is not a variable, so there can be no correlation between this theory and traditional villages.
Lines 527-532. The reader should understand the conclusions without reading the results. The paths are quite interesting and should not be left here without development. In fact, I believe the conclusions should focus primarily on the paths, at least the first two, which have the higher explanatory power.
Lines 533-545: Are these recommendations? These should not be listed as conclusion (3).
Author Response
Response to Reviewer 1 Comments
Thanks: The co-authors and I would like to thank you for the time and effort spent in reviewing the manuscript. According to your advice, we amended the relevant part in manuscript. All of your questions were answered one-by-one.
Point 1: Lines 17-21: The take-home messages of the research should be clearly stated. I believe the paths should be mentioned here.
Response 1: Thank you for your valuable comments, according to your suggestions to complete the revision.
Point 2: Lines 25-35. This introductory paragraph needs to be placed in lines of research, with appropriate citations.
Response 2: Thank you for your suggestion, what we want to express in this paragraph is that traditional villages are a unique settlement situation in China, and how they have changed in the past 40 years due to the impact of urbanization, we think this place is about the current context of traditional villages and the risks and challenges they are facing, after careful consideration we have kept this place unchanged
Point 3: Line 49: The acronym QCA needs to be explained here.
Response 3: Thank you very much for your careful review, the acronym QCA has been explained as necessary in the article.
Point 4: Lines 61-64. These affirmations need citations to support it. Please cite key references.
Response 4: Thank you for your suggestion, the necessary references have been added as needed to cite.
Point 5: Line 68: Why are villages more vulnerable to uncertainties than cities?
Response 5: Thank you for your valuable question. We believe that, compared to the countryside, cities have better systems, management and the ability to deal with emergencies, while traditional villages are mostly on the periphery of cities, and their geographical and transportation environments are lagging behind those of cities. In addition, traditional villages have been affected by urbanization and other factors, and their protection and development have been affected in many ways. Especially for the case studied in this paper, the traditional villages in Lishui City are located in mountainous areas and are affected by natural disasters such as typhoons, floods and mudslides all year round, so the survival of traditional villages is threatened to a certain extent. Secondly, the population in traditional villages seeks for vitality, which leads to a large rural exodus and the gradual hollowing out of traditional villages, which in turn leads to the increasing decay of village environment and architecture, and the conservation of traditional villages faces certain challenges and the vulnerability of villages is gradually strengthened. Therefore, we believe that villages are more vulnerable to uncertainty factors than cities.
Point 6: Line 91: What kind of factors?
Response 6: We are sorry for the error, there is a language problem in the text, we have modified it.
Point 7: Line 120: Use of quotation marks is awkward. Does this refer to the model you have developed? Maybe you can mention that it is a model.
Response 7: We are sorry for the error, there is a language problem in the text, we have revised it according to your comments to enhance readability.
Point8: Lines 394-479: This is really interesting, but is quite hidden in the manuscript.
Response 8: Based on your valuable comments, we decided to add "7.1 Traditional Village Protection and Development Strategies" to the discussion section to enrich the content of the article and form a more intuitive guide to the protection and development of traditional villages.
Point 9: Lines 480-496: The discussion needs to compare the results of the present research with those of previous published research.
Response 9: Revisions have been completed based on your suggestions.
Point 10: Lines 498-505: The conclusions section should focus on summarizing the results, not arguing on theoretical issues.
Response 10: Revisions have been completed based on your suggestions.
Point 11: Line 514: Resilience theory is not a variable, so there can be no correlation between this theory and traditional villages.
Response 11: The resilience theory emphasizes the system's ability to resist, adapt and transform when disturbed by the outside world, which is suitable for the study of traditional village conservation and development, and provides a new direction and path for the conservation and development of traditional villages. The paper also focuses on the role of several resilience elements for traditional village conservation and development, therefore, we believe that there is a strong correlation between resilience theory and traditional village conservation and development, and produce a corresponding correlation framework to further explain that resilience theory can effectively improve the risk coping capacity of traditional villages. Finally, we have rephrased this in accordance with your comments, so as to express our views more visually.
Point 12: Lines 527-532. The reader should understand the conclusions without reading the results. The paths are quite interesting and should not be left here without development. In fact, I believe the conclusions should focus primarily on the paths, at least the first two, which have the higher explanatory power.
Response 12: Based on your valuable comments, we have expanded the descriptions of the 4 paths in the '7.1 Traditional Village Conservation and Development Strategies' section and the conclusion section respectively to enrich the content of the article.
Point 13: Lines 533-545: Are these recommendations? These should not be listed as conclusion (3).
Response 13: According to the calculation results, the univariate variable of landscape environmental protection (JGHJ) has a consistency of more than 0.9 (i.e., 0.930789), which theoretically indicates that this variable constitutes a necessary condition in the conservation and development of traditional villages, and at the same time, for traditional villages, the conservation of landscape environment has a significant influence on the overall conservation and development of the villages. Second, the variable of village industrial development (CLCY) exists as a core condition in multiple histories paths. Therefore, combining the above results and the actual situation, this paper determines the two as the foundation and guarantee of traditional village conservation and development, respectively.
Thank you again for your time and effort in reviewing the manuscript. Best wishes for you!

Reviewer 2 Report
· In the abstract there could be the research aim
· what is the practical relevance of the study especially in the selected villages???
· It is mentioned that empirical research was conducted on 28 national-level traditional villages in Liandu District and Qingyuan County, Lishui City-why this City is included, what rural elements it has?
· There are no details about the interview (e.g. questions, informants selection criteria, etc.)
· In the Discussion part there is recommended to see the results interconnection with previous research findings.
· How this example, research findings can contribute to other countries village traditional village protection and development from the perspective of resilience theory? Can it be easily adopted to other countries?
Author Response
Response to Reviewer 2 Comments
Thanks: The co-authors and I would like to thank you for the time and effort spent in reviewing the manuscript. According to your advice, we amended the relevant part in manuscript. All of your questions were answered one-by-one.
Point1: In the abstract there could be the research aim
Response 1: Thank you for your suggestions, we have improved the abstract section, see the revised manuscript for details.
Point2: what is the practical relevance of the study especially in the selected villages???
Response 2: The main research significance of the article is to explore the role of different influencing factors on the protection and development of traditional villages on the basis of resilience theory, and to explore the high-level path of traditional villages protection and development with the help of fuzzy set qualitative comparison analysis, so as to help the sustainable development of traditional villages. The 28 villages selected in this paper are the national traditional villages in Lishui city that are listed in the Protection List of Chinese Traditional Villages. In addition, Lishui city is a typical representative of the mountainous area in southwest Zhejiang, and the traditional villages are vulnerable to natural disasters and other impacts, and the regional socio-economic development level of Lishui city is relatively low, and its traditional villages are in greater need of protection and development.
Point3: It is mentioned that empirical research was conducted on 28 national-level traditional villages in Liandu District and Qingyuan County, Lishui City-why this City is included, what rural elements it has?
Response 3: Lishui City is a typical representative of the mountainous area in southwest Zhejiang, and most of its traditional villages still retain their original features. Among them, Liandu District, as the administrative center of Lishui City, can intuitively reflect the impact and influence of urbanization on suburban traditional villages; Qingyuan County, as the easternmost district of Lishui City and the border between Zhejiang and Fujian Provinces, its traditional villages are mostly located in relatively remote areas with backward transportation and economic conditions, so they are obviously comparable. In conclusion, the study on the protection and development of traditional villages in Lishui City has a certain representativeness.
Point4: There are no details about the interview (e.g. questions, informants selection criteria, etc.)
Response 4: Due to space limitations, the specific content of the interviews is therefore not detailed in the text. An appendix has been added at the end of the article to show the contents of the specific research interviews in detail (As shown in the table below), and your valuable comments are greatly appreciated.
Interview Outline
Place of interview |
|
Date of interview |
|
Recorder |
|
||
Interview subject and background |
|
||||||
Interview content |
|||||||
Interview questions |
Answer records |
Remarks |
|||||
Questions regarding natural disaster response. (1) Are they affected by natural disasters? What is the type? (2) Has the construction of infrastructure and public services been completed (e.g., electricity and telecommunications, water supply and drainage, road laying, etc.)? (3) Are there enough emergency shelters and emergency measures available? (4) Is disaster prevention and safety education regularly disseminated?
Questions about landscape protection. (1) Are there any historical references to the spatial formation of the village? How is it preserved? (2) Is the landscape environment in the village subject to human damage? (3) How about the protection of historical and cultural buildings? Will the knowledge of traditional village protection be regularly promoted? (4) What are the native and iconic tree and flower species? How is the preservation? (5)Does the phenomenon of private construction affect the overall landscape pattern of the village? (6) Is the necessary environmental health improvement and upgrading work carried out (e.g. road hardening, lighting, public toilets, sewage disposal, etc.)?
Questions about the governance of social organizations. (1) What is the resident population in the village? What is the age structure? (2) Village participation in public affairs in the village? (3) Are there any organizations or institutions to resolve disputes between villages and maintain social order? Are there any village rules and regulations as well as policy systems? (4) How about mutual help relationship among neighbors? (5) Are school-age education as well as health and medical services being met? (6) Are there outside organizations involved in village management and construction? (7) Are there platforms and resources for young people to return to their hometowns to start their own businesses?
Questions regarding the transmission of cultural elements. (1) What are the traditional culture and customs in the village? Are they still surviving? (2) Is the village carrying out traditional culture transmission and protection work? Are there cultural inheritors? Are there any necessary management and protection measures? (3) Is there any intangible cultural heritage (hand weaving, food, songs and dances, etc.)? (4) Are cultural industries (cultural creation, photography, etc.) developed using the cultural elements of the village? (5) How much do villagers participate in and identify with traditional culture and customs?
Questions about the industrial development of the village. (1) What are the main industries or special industries in the village? (2) What is the percentage of agricultural income? Is there any necessary transformation and development? (3) How diversified are the industries in the village? Is there any integrated development according to the characteristics of the village, cultural resources, natural resources combined with tourism industry, cultural industry, etc.? (4) Local employment opportunities? |
|
|
|||||
Point5: In the Discussion part there is recommended to see the results interconnection with previous research findings.
Response 5: Thank you very much for your valuable suggestions we will continue to deepen the concluding part of the article and enhance the correlation with the previous calculations.
Point6: How this example, research findings can contribute to other countries village traditional village protection and development from the perspective of resilience theory? Can it be easily adopted to other countries?
Response 6: As a new way of thinking about risk governance, resilience theory has gradually become a new topic of widespread interest among scholars at home and abroad, and the theory is gradually maturing and being applied. In this paper, mainly on the basis of resilience theory, combined with fuzzy set qualitative comparative analysis method, we explore 4 high level grouping paths of traditional village protection and development. According to your valuable opinions, we decided to add '7.1 Traditional Village Conservation and Development Strategy' to the discussion section, so as to enrich the article content and form a more intuitive guidance of traditional village conservation and development.
Thank you again for your time and effort in reviewing the manuscript. Best wishes for you!

Reviewer 3 Report
The manuscript topic is ´ Influencing Factors of Traditional Village Protection and Development from the Perspective of Resilience Theory´. The main research aim was to explore the protection and development of traditional villages from the perspective of resilience theory.
The manuscript’s strengths:
The general approach of the manuscript is especially good. The manuscript is informative and well structured. The title matches the content. The topic fits the scope of Land journal as well as the section (Land Planning and Architecture) and the case is relevant. The introduction and literature review include sufficient references. The conclusions match the research idea.
The manuscript’s weaknesses:
1) Table 1: You should remove the duplicate text (Ecological resilience).
2) Section 6.1: The theoretical part of the section is to be moved to the Methodology section.
3) Line 390: There is a reference to a missing Table 13.
4) The reference list should be extended.
Generally, after revision, the paper can be accepted for publication.
Author Response
Response to Reviewer 3 Comments
Thanks: The co-authors and I would like to thank you for the time and effort spent in reviewing the manuscript. According to your advice, we amended the relevant part in manuscript. All of your questions were answered one-by-one.
Point 1: Table 1: You should remove the duplicate text (Ecological resilience).
Response 1: Thank you very much for your careful review, the information in Table 1 has been revised and replaced with the relevant content on economic resilience.
Point 2: Section 6.1: The theoretical part of the section is to be moved to the Methodology section.
Response 2: This section is only one point in the methodology. The reason we set it up this way is to enhance the readability of the article and to make it easier for readers who do not know the methodology to understand more intuitively what the article wants to say there, and after careful consideration by the author of the article, we hope to keep the structure of the section unchanged.
Point 3: Line 390: There is a reference to a missing Table 13.
Response 3: Sorry for the error, the 'Table 13' expression has been removed.
Point 4: The reference list should be extended.
Response 4: References have been added as appropriate to the actual needs of the article.
Thank you again for your time and effort in reviewing the manuscript. Best wishes for you!

Reviewer 4 Report
Thank you very much for the opportunity to read this paper. This is interesting research on complex issues about the countryside and in the context of resilience. However, I have a few comments on this.
1. The article has a good methodological basis. Many theoretical issues remain poorly done. The literature review is very poor. The problem should be better placed in the theories of sustainable development, resilience and territorial development.
2. I do not see a clearly defined and developed research goal.
3. Therefore, the discussion is very weak because there is not good theoretical basis in the introduction.
4. Better reference should be made to studies conducted in other countries, especially non-Asian models (earlier processes).
Author Response
Response to Reviewer 4 Comments
Thanks: The co-authors and I would like to thank you for the time and effort spent in reviewing the manuscript. According to your advice, we amended the relevant part in manuscript. All of your questions were answered one-by-one.
Point 1: The article has a good methodological basis. Many theoretical issues remain poorly done. The literature review is very poor. The problem should be better placed in the theories of sustainable development, resilience and territorial development.
Response 1: For reasons of space, the previous version of the article did not focus on the theoretical background part, therefore, according to your suggestion, this revision adds the research progress on the resilience theory and other contents.
Point 2: I do not see a clearly defined and developed research goal.
Response 2: We have already added improvements according to your suggestions.
Point 3: Therefore, the discussion is very weak because there is not good theoretical basis in the introduction.
Response 3: Based on your valuable comments, we decided to add "7.1 Traditional Village Protection and Development Strategies" to the discussion section to enrich the content of the article and form a more intuitive guide to the protection and development of traditional villages.
Point 4: Better reference should be made to studies conducted in other countries, especially non-Asian models (earlier processes).
Response 4: The necessary references have been added according to your suggestions.
Thank you again for your time and effort in reviewing the manuscript. Best wishes for you!

Round 2
Reviewer 4 Report
Good job.